# Assessing machine learning for fair prediction of ADHD in school pupils using a retrospective cohort study of linked education and healthcare data

Lucile Ter-Minassian,[1] Natalia Viani,[1] Alice Wickersham [ID],[1] Lauren Cross,[1] Robert Stewart,[1,2] Sumithra Velupillai,[1] Johnny Downs [ID] [2,3]

NV, AW and LC contributed equally.

¹Department of Psychological Medicine, King's College London, London, UK
²South London and Maudsley NHS Foundation Trust, London, UK
³Department of Child and Adolescent Psychiatry, King's College London, London, UK

**Correspondence to**
Dr Johnny Downs;
johnny.downs@kcl.ac.uk

## ABSTRACT

**Objectives** Attention deficit hyperactivity disorder (ADHD) is a prevalent childhood disorder, but often goes unrecognised and untreated. To improve access to services, accurate predictions of populations at high risk of ADHD are needed for effective resource allocation. Using a unique linked health and education data resource, we examined how machine learning (ML) approaches can predict risk of ADHD.

**Design** Retrospective population cohort study.

**Setting** South London (2007–2013).

**Participants** n=56 258 pupils with linked education and health data.

**Primary outcome measures** Using area under the curve (AUC), we compared the predictive accuracy of four ML models and one neural network for ADHD diagnosis. Ethnic group and language biases were weighted using a fair pre-processing algorithm.

**Results** Random forest and logistic regression prediction models provided the highest predictive accuracy for ADHD in population samples (AUC 0.86 and 0.86, respectively) and clinical samples (AUC 0.72 and 0.70). Precision-recall curve analyses were less favourable. Sociodemographic biases were effectively reduced by a fair pre-processing algorithm without loss of accuracy.

**Conclusions** ML approaches using linked routinely collected education and health data offer accurate, low-cost and scalable prediction models of ADHD. These approaches could help identify areas of need and inform resource allocation. Introducing 'fairness weighting' attenuates some sociodemographic biases which would otherwise underestimate ADHD risk within minority groups.

## STRENGTHS AND LIMITATIONS OF THIS STUDY

⇒ We leveraged a large and innovative data linkage between routinely collected health and school data.
⇒ Machine learning approaches were applied to predict attention deficit hyperactivity disorder in a population and clinical cohort.
⇒ Models were weighted to account for ethnic group and language biases.
⇒ Further validation is needed to establish the stability and generalisability of these models.

substantial diagnostic delays preclude effective early intervention.[6] It is therefore vital that adequate resources are made available for timely ADHD detection, diagnosis and intervention. To effectively allocate adequate resources for ADHD, commissioners need accurate information on the level of need in schools, services and geographical regions.

Pooled estimates from population surveys suggest that 5.3% of the global population meet criteria for ADHD.[2] This could be taken as a broad indicator of level of need in any given population, but some cases identified via population surveys will not be sufficiently severe or complex to require input from clinical services. Therefore, prevalence estimates from population surveys likely overestimate the true level of need in the population. Alternatively, level of need could be estimated from the prevalence of ADHD cases presenting to clinical services. However, only a fraction of cases requiring input seek help, receive a referral, diagnosis and treatment from clinical services.[7] Therefore, prevalence estimates from clinical services likely underestimate the true level of need in the population.

Overall, there remains a need for accurate indicators of level of need which can inform commissioning and resource distribution. At the school level, mental health screening

## INTRODUCTION

Attention deficit hyperactivity disorder (ADHD) is a neurodevelopmental disorder characterised by a persistent pattern of inattention, hyperactivity and impulsive behaviour.[1] Although symptoms emerge in preschool, ADHD tends to lead to clinical levels of impairment during the first few years of statutory education.[2] ADHD can lead to poor outcomes across social, educational and health domains.[3 4] It can be treated,[5] but

questionnaires administered by teachers can be used to identify pupils at risk of ADHD, which in could turn provide an estimate of ADHD prevalence for that school.[8] However, this approach is resource intensive, and cannot practically be scaled up to larger populations to estimate burden of need in a wider geographical area.[9] Machine learning techniques may be able to predict ADHD diagnosis at the individual level, but existing approaches again use data sources which cannot be practically scaled up to larger populations, such as MRI and electroencephalography data.[10–13] Such models might also be prone to systematic biases[14]: ADHD is underdiagnosed and undertreated among ethnic minority groups, who are therefore unlikely to be adequately represented in the data on which such models are built, unless efforts are made to adjust for such biases.[15 16]

Therefore, there remains a need for accurate and unbiased ADHD prediction methods which are scalable to large populations or geographical regions. Administrative pupil-level data are already routinely collected and curated by most schools, and are available for whole populations.[17] If machine learning approaches can be effectively applied to these data to build validated and unbiased models predicting childhood ADHD, they could be used to estimate burden of need and guide resource allocation at area, school and service levels. In this study, we applied machine learning approaches to a large, linked administrative school and clinical dataset to: (1) classify pupils in the population with a high likelihood of clinical ADHD, (2) differentiate pupils with ADHD from pupils with other clinical disorders, (3) explore the contribution of demographic, socioeconomic and education features to an ADHD diagnosis, and (4) evaluate a potential method to reduce sociodemographic biases.

## METHODS
### Study design
This was a retrospective cohort study, using an existing data linkage between the National Pupil Database (NPD) and South London and Maudsley National Health Service Foundation Trust Child and Adolescent Mental Health Services (SLaM CAMHS) (online supplemental table S1).[18 19]

SLaM is one of Europe's largest providers of mental healthcare, serving the London boroughs of Croydon, Lambeth, Lewisham and Southwark. SLaM is the monopoly provider of local CAMHS services, including ADHD diagnostic services.[18] Anonymised individual-level clinical data for these services are accessible for research via the Clinical Record Interactive Search (CRIS).

CRIS data for SLaM CAMHS services have been linked at the individual level to the NPD. The NPD is a longitudinal database of pupil-level and school-level data.[17] The NPD pupil census is collected annually and contains a snapshot of pupils attending state-maintained schools in England, while NPD attainment datasets hold data

for all statutory assessments that pupils complete during primary and secondary education.

The CRIS-NPD linkage process has been described in detail elsewhere.[18 19] Linkage was conducted for referrals to SLaM between 2007 and 2013, for children and adolescents aged 4–18 years. Fuzzy deterministic matching was conducted based on personal identifiers in CRIS, which were sent to the Department for Education for linkage to the NPD. The match rate was 82.5%. As a result of this data linkage, we had access to educational records for pupils who were referred to SLaM from both inside and outside the catchment area, and local pupils who were not referred to SLaM.

### Patient and public involvement
The health and education data linkage used in this study was developed in consultation with several clinical, patient and caregiver groups via the SLaM Biomedical Research Centre patient engagement programme.[19] A lay summary of the data linkage can be viewed at https://www.maudsleybrc.nihr.ac.uk/facilities/clinical-record-interactive-search-cris/cris-data-linkages/.

### Participants
The analysed sample comprised children aged between 6 and 7 years, who were enrolled in mainstream state educational services (not specialist or alternative educational facilities), and who had education and attainment characteristics as captured at the Early Years Foundation Stage Profile (EYFSP; typically aged 4–5 years) and at Key Stage 1 (KS1; typically aged 6–7 years). Pupils were included if they were resident of the SLaM catchment area from 1 September 2007 to 31 August 2013. A resulting n=57 149 pupils were eligible for inclusion.

### Main outcome
Individual-level diagnosis of ADHD was measured from both structured and unstructured diagnosis fields in CRIS between September 2007 and August 2013. In structured fields, ADHD diagnosis was defined as International Classification of Diseases 10th Revision codes F90.0, F90.1, F90.8 or F90.9 (hyperkinetic, other hyperkinetic disorders, attention deficit disorder, hyperkinetic conduct disorder).[20] In unstructured fields, ADHD diagnosis was identified and manually evaluated using previously established natural language processing techniques (positive predictive value ~0.82).[21–24]

### Educational features and other covariates
School performance indicators were collected from EYFSP and KS1 datasets in the NPD. The EYFSP comprises teacher-assessed social, linguistic, physical and cognitive development at the end of the first year of schooling.[25] KS1 comprises teacher-assessed English, maths and science at the end of the third year of formal education.[26] Gender, age, ethnicity, free school meal eligibility (eFSM; a proxy measure for deprivation), attendance, first language, special educational need (SEN) and looked after child (LAC) status were also measured from the NPD. In total,

45 characteristics were included (online supplemental tables S2 and S3).

## Predictive model preparation and building

To minimise the potential for reverse causality, we excluded pupils who were diagnosed with ADHD prior to their KS1 assessment (n=77; 10.5% of ADHD cases). We used a complete case approach, excluding pupils with missing data from our eligible sample (n=814; 1.4%). The resulting n=56 258 formed our population cohort.

To develop and compare machine learning classifiers, the dataset was randomly divided into a training set (n=42 192; 75.0%), validation set (n=7033; 12.5%) and test set (n=7033; 12.5%) with a similar proportion of ADHD cases (~1%) in each set. The training set was used to train the algorithms, while the validation set was used for model refinement (eg, hyperparameter tuning) and comparison. Final performance was evaluated on the test set. We compared several machine learning algorithms to classic statistical logistic regression (LR) approaches, and also a supervised approach based on neural networks, using a multilayer perceptron (MLP). The following classifiers were compared: LR, random forest (RF), support vector machines (SVM), Gaussian Naive Bayes (GNB) and the MLP. For GNB, we first tested that continuous features associated with each class were distributed according to a Gaussian distribution. For the neural network, we used a four-layer architecture (input layer, two hidden layers, output layer).[27]

## Selection procedure for machine learning algorithms

Model performances were measured with the area under the receiver operator characteristics curve (AUC). We also report precision, recall and F1-score with a fixed classification threshold of 0.5. The F1-score was calculated using the 'weighted' option, which calculates metrics for each label and finds their average weighted by support. For each given value, a 95% CI of the outcome measure was estimated by bootstrapping 100 times the subjects in the set. We also examined the precision-recall curve (PRC) with precision (y-axis) and recall (x-axis) plotted for different probability thresholds. All the experiments with machine learning algorithms were conducted in Python with functions from the scikit learn library for classification (V.0.20.3).[28] The MLP was implemented using the Keras framework (V.2.2.4).

## Tuning, resampling and ensemble methods for classification

For LR and RF, tuning of the hyperparameters was made using grid search with 10-fold cross-validation on the validation set (full details available by request to the author). The GNB classifier had no parameters to tune. Tuning of SVM was done empirically (online supplemental table S4). To compensate for class imbalance of ADHD outcomes, we compared two different methods: random subsampling and Tomek links.[29] For random subsampling, we successively fitted our model to multiple subsamples. Each sample contained all ADHD cases and 1000 subjects without any ADHD diagnosis. We calculated the AUC for each fitted model and took the average value of AUC. For Tomek links, we used two approaches—removing all links from the dataset and removing the majority class samples that were part of a Tomek link. In an attempt to improve classification performance, ensemble methods were used on selected machine learning methods.

## Selection and testing the final models

Two criteria were defined for the final models' selection: performance and transparency, that is, the models' ability to determine feature contribution, which enabled examination of potential biases inherent to the data resources used. High-performing models were considered a priori as having AUC ≥0.85 on the validation set.

Models which were both transparent and performed well were tested for diagnostic efficiency, the purpose being to assess whether the predictive performance from the population sample could generalise to clinical populations. To test the specificity of the best models for ADHD from other clinical mental health problems, we repeated our approach on a purely clinical cohort containing only the children who were present in the CRIS dataset, thus containing samples of children who presented with ADHD and non-ADHD diagnoses. Once all models were tested on the full set of features, we then refined the models to improve accuracy, reduce overfitting and increase potential for translation into busy clinical settings—making the model less vulnerable to missing information. To identify the most important features for LR models, we used beta coefficients, while RF feature importance represents the decrease in node impurity on a given branch within a decision tree, weighted by the probability of reaching that node.

## Bias reduction

We investigated bias reduction for the LR and RF models trained on the population cohort using the AI Fairness 360 toolkit.[30] We used a pre-processing algorithm as we could access and modify the data directly and the data were inherently biased due to unfair clinical case ascertainment. Pre-processing algorithms learn a new representation based on protected attributes, that is, features that partition the dataset into groups that should have equal probability of getting a favourable label.[31] In our case, protected attributes were represented by ethnicity and language status, while the favourable label is an ADHD diagnosis. The reweighting algorithm was chosen for its ability to treat two protected attributes at the same time and for the transparency of its fairness process.[32] This algorithm modified the dataset by generating different weights for the training examples in each (group, label) combination, to ensure fairness before classification.

Online supplemental figure S1 displays the pre-processing fairness pipeline. We defined white and English speaking as the 'privileged' group, and non-white and non-English speaking as the 'unprivileged' group, thus obtaining two non-overlapping groups to be

weighted differently. Other combinations (non-white and English speaking, white and non-English speaking) were not explicitly reweighted.

## RESULTS
### Cohort characteristics
In total, n=652 pupils in our sample were diagnosed with ADHD (1.16% of the population cohort; 15.6% of the clinical cohort) (table 1). In the population cohort, gender was evenly balanced, and pupils were predominantly black (n=22 904, 40.71%) and spoke English as a first language (n=37 432, 66.54%). However, males were over-represented in the clinical cohort (n=2848, 68.17%), as were white ethnic groups (n=1712, 40.98%) and English first language speakers (n=3390, 81.14%). This is consistent with national surveys which report greater psychiatric morbidity among boys aged 5–10 years, and higher rates of help-seeking and detection among white, English-speaking groups.[33 34] Compared with the population cohort, the clinical cohort also had a lower EYFSP and KS1 attainment, lower attendance, and higher rates of exclusion, eFSM, LAC and SEN.

### Model performance: identifying ADHD from population cohort
Once all models were tested on the full set of 68 features, we refined the models to improve accuracy, reduce overfitting and increase potential for translation into busy clinical settings. Table 2 displays AUC, precision, recall and F1-score, including confidence parameters for all tuned models on both validation and test sets, for classifying ADHD from the population cohort (figure 1 for receiver operator characteristics (ROC) analysis). Hyperparameter tuning improved RF significantly. Precision was high (above 0.985 on both the validation and test set) for LR, RF and GNB. LR, RF and the MLP all had an AUC ≥0.85 on both the validation and the test set. SVM did not discriminate ADHD diagnosis as well (AUC <0.82). Overall, LR and RF were the highest performing and transparent models.

### Model performance: identifying ADHD from clinical cohort
Table 2 displays precision, recall, F1-score and final hyperparameters trained, validated and tested on the clinical cohort. Precision was ~0.80 for ADHD on both the validation and test set for both models on the clinical cohort. Both models' overall predictive performance (AUC 0.70) was much lower relative to their performance on the population cohort.

Resampling methods used to deal with class imbalance did not improve the performances significantly. Overall, random subsampling performed better than Tomek links. As for ensemble approaches, a slight improvement in performance was noticed for RF only, while LR obtained better performance when using resampling methods. The PRCs for both population and clinical cohorts are displayed in online supplemental figures S2 and S3. The models show significantly lower performance than ROC curve results, although they do demonstrate greater performance than a random classifier, based on PRC metrics as summarised in online supplemental table S5 comparing the PRC and ROC for LR, RF and random classification.

### Bias reduction using English language and white ethnicity as protective attributes
Table 3 displays results on the population cohort (test set) when using the LR and RF models trained on transformed data (ie, with instance weights from the fairness reweighting algorithm). The disparate impact score displays the ratio of the probability of a favourable outcome for unprivileged instances and the probability of a favourable outcome for privileged instances (close to 1 in a fair dataset). The reweighting algorithm obtained a considerable improvement of the disparate impact (from 0.15-0.60 to 1), without harming the models' performance on ADHD classification: for the LR/RF models, the AUC remained above 0.880/0.858 in all cases.

### Feature contribution for ADHD prediction in population and clinical cohorts
Table 4 displays the most important features overall, plus ethnicity, for LR and RF models in the population and clinical cohorts (all features in online supplemental table S6). KS1 writing performance was a strongly protective/discriminant factor against ADHD. Gender, KS1 attendance, SEN status, and personal, social and emotional development were also strongly discriminant features, while the importance of English language and ethnicity was reduced following reweighting.

## DISCUSSION
This study aimed to predict pupils with a high likelihood of ADHD diagnosis using data available at the population level. To our knowledge, it is the first study to apply machine learning approaches to large-scale, routinely collected linked health and education data. These innovative methods show that ADHD in general population and clinical cohorts can be predicted with high levels of accuracy. Promisingly, this level of predictive accuracy is comparable with traditional survey-based screening methods in ADHD, and with other studies using routinely collected data to predict health outcomes.[35–38]

These findings suggest that machine learning approaches can accurately predict ADHD in large general population and clinical cohorts using existing and widely available administrative data. These models may therefore be suited to provide health intelligence to local policymakers at low cost, and inform decisions on the allocation of disorder-specific resources at an area, school or service level. Tentatively, these models may also offer a means to predict ADHD at the individual level, although greater precision would be desirable for this purpose—this is an area for future work.

**Table 1** Characteristics of the population cohort and clinical cohort

| | Population cohort n=56 258 | | Clinical cohort n=4178 | |
|---|---|---|---|---|
| ADHD (n, %) | 652 | 1.16 | 652 | 15.61 |
| Gender (male) (n, %) | 28 434 | 50.54 | 2848 | 68.17 |
| Summer birth (n, %) | 19 274 | 34.26 | 1443 | 34.53 |
| English as first language (n, %) | 37 432 | 66.54 | 3390 | 81.14 |
| Ethnicity (n, %) | | | | |
| Asian | 4833 | 8.59 | 129 | 3.09 |
| Black | 22 904 | 40.71 | 1461 | 34.97 |
| Chinese | 484 | 0.86 | 10 | 0.24 |
| White | 18 218 | 32.38 | 1712 | 40.98 |
| Mixed | 7116 | 12.65 | 667 | 15.97 |
| Other | 2702 | 4.80 | 189 | 4.52 |
| EYFSP attainment (range; mean (SD)) | | | | |
| Personal, social and emotional development | 0.28–6.60 | 4.98 (0.99) | 0.28–6.60 | 4.32 (1.20) |
| Communication, language and literacy | 1.31–6.67 | 4.97 (0.99) | 1.31–6.67 | 4.39 (1.16) |
| Problem-solving, reasoning and numeracy | 0.86–6.54 | 4.98 (0.99) | 0.86–6.54 | 4.46 (1.27) |
| Knowledge and understanding of the world | 0.95–6.58 | 4.97 (0.99) | 0.95–6.58 | 4.54 (1.23) |
| Physical development | −0.01 to 6.43 | 4.98 (0.99) | −0.01 to 6.43 | 4.45 (1.31) |
| Creative development | 0.58–6.76 | 4.97 (0.99) | 0.58–6.76 | 4.50 (1.25) |
| KS1 attainment (range; mean (SD)) | | | | |
| Maths | 0–27 | 15.50 (4.10) | 3–27 | 13.24 (4.57) |
| Reading | 0–27 | 15.66 (3.88) | 3–27 | 13.66 (5.02) |
| Writing | 0–27 | 14.31 (3.71) | 3–27 | 11.61 (4.58) |
| Science | 0–27 | 15.23 (3.55) | 3–21 | 13.29 (4.43) |

| | EYFSP | | KS1 | | EYFSP | | KS1 | |
|---|---|---|---|---|---|---|---|---|
| | n | % | n | % | n | % | n | % |
| eFSM (n, %) | 18 175 | 32.31 | 18 177 | 32.31 | 1880 | 44.50 | 1882 | 45.05 |
| Looked after child status (n, %) | 98 | 0.17 | 202 | 0.36 | <10 | <10 | 103 | 2.47 |
| Mainstream schooling (n, %) | – | – | 55 854 | 99.28 | – | – | 4006 | 95.88 |
| Exclusion (n, %) | – | – | 690 | 1.23 | – | – | 494 | 11.82 |
| No SEN (n, %) | 45 113 | 80.19 | 45 116 | 80.20 | 2031 | 48.61 | 2031 | 48.61 |
| Statement (n, %) | 924 | 1.64 | 922 | 1.64 | 356 | 8.52 | 356 | 8.52 |
| SEN school action (n, %) | 5766 | 10.25 | 5766 | 10.25 | 675 | 16.16 | 675 | 16.16 |
| SEN school action plus (n, %) | 4454 | 7.92 | 4453 | 7.92 | 1116 | 26.71 | 1116 | 26.71 |
| SEN behavioural, social and emotional (n, %) | 331 | 0.6 | 331 | 0.6 | <10 | <10 | <10 | <10 |

ADHD, attention deficit hyperactivity disorder; eFSM, free school meal eligibility; EYFSP, Early Years Foundation Stage Profile; KS1, Key Stage 1; SEN, special educational need.

We also demonstrate that sociodemographic biases inherent in the data and algorithms can be reduced with a fairness reweighting step to ensure that pupils from certain backgrounds are not systematically overlooked as being at risk of ADHD. In a geographical area with a large proportion of people from ethnic minority and non-English-speaking backgrounds, minimising such biases is crucial for accurately predicting need for ADHD provision in the population.[24]

**Table 2** Comparison of models: (1) predictive performance of all trained models on the population cohort; (2) performance of LR and RF in identifying ADHD from other child-onset psychiatric disorders within the clinical cohort (median and 95% CI)

| Models | AUC | | Precision | | Recall | | F1-score | |
|---|---|---|---|---|---|---|---|---|
| | Validation*† | Test*† | Validation*† | Test*† | Validation*† | Test*† | Validation*† | Test*† |
| **Population cohort** | | | | | | | | |
| LR | 0.862 (0.840 to 0.874) | 0.900 | 0.985 (0.985 to 0.986) | 0.987 | 0.835 (0.822 to 0.848) | 0.827 | 0.900 (0.892 to 0.908) | 0.895 |
| RF | 0.857 (0.842 to 0.869) | 0.860 | 0.985 (0.985 to 0.986) | 0.986 | 0.816 (0.802 to 0.833) | 0.802 | 0.889 (0.880 to 0.899) | 0.880 |
| SVM | 0.811 (0.793 to 0.833) | 0.830 | 0.980 (0.979 to 0.981) | 0.981 | 0.936 (0.930 to 0.943) | 0.908 | 0.957 (0.954 to 0.96) | 0.942 |
| GNB | 0.855 (0.846 to 0.873) | 0.849 | 0.985 (0.985 to 0.986) | 0.987 | 0.835 (0.822 to 0.850) | 0.707 | 0.890 (0.882 to 0.909) | 0.817 |
| MLP | 0.855 (0.834 to 0.864) | 0.883 | 0.978 (0.975 to 0.987) | 0.977 | 0.988 (0.982 to 0.990) | 0.988 | 0.983 (0.979 to 0.985) | 0.983 |
| **Clinical cohort** | | | | | | | | |
| LR | 0.718 (0.701 to 0.735) | 0.694 | 0.813 (0.804 to 0.834) | 0.797 | 0.659 (0.647 to 0.673) | 0.646 | 0.705 (0.694 to 0.717) | 0.693 |
| RF | 0.699 (0.682 to 0.71) | 0.689 | 0.808 (0.784 to 0.823) | 0.808 | 0.639 (0.613 to 0.672) | 0.808 | 0.689 (0.665 to 0.715) | 0.654 |

*ADHD cases: n=82.
†Non-ADHD cases: n=6951 in population cohort, n=441 in clinical cohort.
ADHD, attention deficit hyperactivity disorder; AUC, area under the curve; GNB, Gaussian Naive Bayes; LR, logistic regression; MLP, multilayer perceptron; RF, random forest; SVM, support vector machines.

**Figure 1** ROC curves for the tuned models. LR, logistic regression; MLP, multilayer perceptron; RF, random forest; ROC, receiver operator characteristics; SVM, support vector machines.

This therefore highlights the need to build fairness measures which weight for limited access to diagnosis in certain ethnic and social groups. However, it is important to note that these statistical approaches are unlikely to remove all potential biases inherent to service provision data: for example, parents also influence help-seeking for ADHD, but their characteristics are not fully captured in datasets like the NPD, and so could not be used in reweighting processes.[39] Subset scanning could be a means to detect biases in certain subgroups.[32] It should also be noted that the reweighting algorithm we used was based on the assumption that different ethnic and language groups should have equal probability of a favourable label (ie, ADHD): under these parameters, optimised pre-processing could also have been applied, as it ensures both group fairness and individual fairness, where similar individuals are treated similarly.[40] Further work could explore these methods.

### Strengths and limitations

Our study benefited from a large and novel data linkage capturing almost all pupils from a defined geographical catchment. This reduces the risk of building models on biased and non-representative samples. These approaches are also scalable, as they use information already routinely collected by schools. We used a set of machine learning algorithms that are state-of-the-art techniques and are representative of the wide spectrum of machine learning methods: an ensemble method (RF), a kernel-based method (SVM), a Bayesian method (Naive Bayes) and a neural network (MLP).

However, some limitations should be highlighted. While we sought to address biases in clinical data using reweighting algorithms, biases in the education data (such as inconsistent representation of private or home-schooled pupils) have not been addressed. We also handled missing data using a complete case analysis, which can introduce bias if the missing data are not randomly distributed. An alternative would have been to explore imputation methods; however, given that we compare models of varying complexities (from LR to MLPs), we would have had to adapt the imputation method to the model, resulting in varied datasets. The comparison between the resulting models would therefore have been

**Table 3** Model performance on the reweighted fair dataset resulting from the bias reduction algorithm

| Models | AUC | Precision | Recall | F1-score | DI before | DI after |
|---|---|---|---|---|---|---|
| LR** | 0.880 | 0.986 | 0.810 | 0.885 | 0.60 | ~1.000 |
| RF** | 0.858 | 0.986 | 0.797 | 0.876 | 0.149 | ~1.000 |

*Reweighting model based on the 'privileged' group being both white and speaking English as a first language.
AUC, area under the curve; DI, disparate impact; LR, logistic regression; RF, random forest.

unfair. Encouragingly, the proportion of missing data was very low, and therefore unlikely to materially bias the findings. Nonetheless, further validation of these models is warranted.[41]

We have assumed that diagnostic and treatment discrepancies between different ethnicities and language groups are due to disparity. It could be argued that such discrepancies are in fact reflective of genuine underlying differences in ADHD prevalence in these populations. However, existing evidence suggests that ADHD is underdiagnosed and undertreated in certain ethnic groups; plausible mechanisms along the care-seeking pathway which may result in these disparities have been discussed elsewhere.[15 16]

For reweighting, we dichotomised ethnicity as white/non-white, and language as English/non-English—conducting a more granular reweighting procedure using more precise groupings would be an important area for future research. We also did not conduct reweighting on the basis of any other sociodemographic characteristics such as gender or deprivation, and make no assumptions

as to their role in differential diagnoses—again, these could be areas for future work.

To identify ADHD cases, we used diagnostic codes recorded in structured fields. Evidence suggests that ADHD can be classified from diagnostic codes in electronic health records with a high degree of accuracy; however, such methods of ascertainment are imperfect, and the possibility of diagnostic or administrative misclassification does remain.[42 43]

Replication in other regions or populations would also be important to assess the generalisability of these models, although at present, other educational and clinical data linkages on this scale are scarce. Finally, some of the features used in the analysis had very few observations for the ADHD cases. The potential impact of this is understudied and warrants further investigation.[44 45]

## Conclusions

Overall, this study demonstrates that machine learning approaches using readily available education and clinical data show promise in predicting ADHD in source

**Table 4** Feature contribution for ADHD prediction and the impact of fairness reweighting

| | Population cohort (n=56 257) | | | | Clinical cohort (n=4178) | |
|---|---|---|---|---|---|---|
| | LR | | RF | | | |
| | Unweighted | Weighted | Unweighted | Weighted | LR | RF |
| KS1 writing score | −0.761 | −0.858 | 0.106 | 0.100 | −0.407 | 0.109 |
| EYFSP personal, social and emotional development | −0.841 | −1.000 | 0.051 | 0.065 | −0.314 | 0.047 |
| KS1 attendance (%) | 0.958 | 1.104 | 0.010 | 0.008 | 0.093 | 0.059 |
| Male gender | 0.608 | 0.626 | 0.029 | 0.061 | 0.389 | 0.142 |
| KS1 no SEN | −0.198 | −0.169 | 0.146 | 0.138 | −0.094 | 0.087 |
| English as first language | 0.696 | 0.183 | 0.049 | 0.006 | 0.113 | 0.032 |
| EYFSP attendance (%) | −0.983 | −1.019 | 0.010 | 0.005 | 0.093 | 0.050 |
| EYFSP problem-solving, reasoning and numeracy | 0.201 | 0.318 | 0.008 | 0.005 | 0.183 | 0.019 |
| White ethnicity | 0.224 | 0.022 | 0.004 | <0.001 | 0.078 | 0.006 |

For the population cohort, coefficients of the model when trained on the reweighted dataset are displayed. Feature importance metrics are different for LR and RF, and not directly comparable. LR beta coefficients represent the log-odds of a given feature, negative values correspond with a decrease in the probability of the case being classified as ADHD and vice versa. RF feature importance is calculated as the decrease in node impurity on a given branch within a decision tree, weighted by the probability of reaching that node. The feature importance value presented represents the average feature importance over all the trees. The table displays the top 5 features for at least one model (marked in bold for the corresponding column) and top 20 features for all four models. The feature 'white ethnicity' was added for comparing its significance after reducing bias. Ranking was made by decreasing significance. For the population cohort, coefficients of both models, when trained on the reweighted dataset using both English and white as protected attributes, are displayed.
ADHD, attention deficit hyperactivity disorder; EYFSP, Early Years Foundation Stage Profile; KS1, Key Stage 1; LR, logistic regression; RF, random forest; SEN, special educational need.

populations. It also highlights that biases inherent in routinely collected data can be mitigated using fair weighting processes. With further validation and replication work, these methods have the potential to estimate burden of need for ADHD provision, thereby informing resource allocation and policy decisions.

**Acknowledgements** The authors are very grateful to Richard White and members of DfE National Pupil Database team, who provided invaluable support throughout the project.

**Contributors** The study was conceived by LT-M, NV and JD. Data extraction was carried out by JD. Data analysis was undertaken by LT-M and NV. Reporting of findings was led by LT-M and NV with support from JD, AW, LC, SV and RS. The guarator for this work is JD. All authors contributed to manuscript preparation and approved the final version.

**Funding** JD is supported by the National Institute for Health Research (NIHR) Clinician Science Fellowship award (CS-2018-18-ST2-014) and has received support from a Medical Research Council Clinical Research Training Fellowship (MR/L017105/1) and Psychiatry Research Trust Peggy Pollak Research Fellowship in Developmental Psychiatry. SV was additionally supported by the Swedish Research Council (2015-00359), Marie Skłodowska Curie Actions, Cofund. RS, LC, JD and SV were additionally supported by a Medical Research Council Mental Health Data Pathfinder Award to King's College London. The Clinical Record Interactive Search (CRIS), RS, JD, NV and SV are part supported by the NIHR Biomedical Research Centre at the South London and Maudsley NHS Foundation Trust and King's College London. RS is additionally part funded by the NIHR Applied Research Collaboration South London (NIHR ARC South London) at King's College Hospital NHS Foundation Trust, and the DATAMIND HDR UK Mental Health Data Hub (MRC grant MR/W014386). AW is in receipt of a PhD studentship funded by the NIHR Biomedical Research Centre at South London and Maudsley NHS Foundation Trust and King's College London. AW is also supported by ADR UK (Administrative Data Research UK), an Economic and Social Research Council (ESRC) investment (part of UK Research and Innovation). [Grant number: ES/W002531/1]

**Patient and public involvement** Patients and/or the public were involved in the design, or conduct, or reporting, or dissemination plans of this research. Refer to the Methods section for further details.

**Patient consent for publication** Not required.

**Ethics approval** The CRIS data resource, including the linked data here, has received research ethics approval for secondary analyses (Oxford REC C, reference 18/SC/0372). CRIS operates an opt-out system, where all patients have the ability to opt out of their anonymised data being used for research. At the time of submission, JD had full ongoing access to the data.

**ORCID iDs**
Alice Wickersham http://orcid.org/0000-0002-7402-7690
Johnny Downs http://orcid.org/0000-0002-8061-295X

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
