## [Reviewer comments · BMJ Open]

ARTICLE DETAILS

TITLE (PROVISIONAL)	Assessing machine learning for fair prediction of ADHD in school pupils using a retrospective cohort study of linked education and healthcare data
AUTHORS	Ter-Minassian, Lucile; Viani, Natalia; Wickersham, Alice; Cross, Lauren; Stewart, Robert; Velupillai, Sumithra; Downs, Johnny

VERSION 1 – REVIEW

REVIEWER	Shi, Yu Mayo Clinic
REVIEW RETURNED	17-Nov-2021

GENERAL COMMENTS	The topic has public health significance. Paper is well written. The authors utilized unique resources of school and clinical datasets. I am not familiar with all the machine learning languages but it seemed to be well performed. I have several questions/comments: 1. It's helpful to present the sensitivity, specificity, PPV, and NPV of the ML models.2. Diagnostic codes on ADHD from clinical charts perform well in identifying true ADHD but are not perfect. It should be mentioned as a limitation. Please see Gruschow 2016 and Shi 2020.3. The work on disparity is very innovative. I don't quite understand if the weighing procedure takes away some of the true differences that are not disparity. For example, male gender is a known risk factor for ADHD diagnosis and probably not a result of discrimination.4. The author acknowledged that the incidence of ADHD is low in the cohort. Can the author compare the incidence to other published studies? Will a higher incidence change the performance of the ML models?
---

REVIEWER	Do, Quan Mayo Clinic Rochester
REVIEW RETURNED	08-Feb-2022

GENERAL COMMENTS	This is an interesting study. I have several comments for the authors. Thank you for the opportunity to review your paper! Your study population was children (6-7 years old), did you have to seek the permission from the parents to involve participants in this study? In total, how many features were used to build the model? Did the authors do data engineering, feature selection, feature ranking to select strong predictors?
--

	Have you checked about outlier/outlier removal? Usually, neural networks (MLP) have better performance than other algorithms, especially Logistic Regression. Do you know why this is not the case in this study? Is your data nonlinear? Missing data (1.4% of data) was deleted. Not sure if you may delete some important cases? There are some imputation methods such as KNN, MICE, or Datawig. Were there any reasons against the use of these imputation methods in this case? It would be better if I can know the changing in accuracy rates through different datasets. Can you please do the 5-fold or 10-fold cross validation and list the differences in accuracy rates of all the used algorithms. Did you test prediction performance with a completely new dataset?
--	---

REVIEWER	Graham, Byron Queens Univ Belfast, Management
REVIEW RETURNED	11-Feb-2022

GENERAL COMMENTS	Thank you for the opportunity to review this paper. There is definitely merit in predicting ADHD at both the individual level and at aggregate levels. I also found the approach to addressing bias in the underlying data to be an interesting approach, and my feeling is that this is an important consideration especially if this type of tool were used to make predictions at the individual level (E.g. to recommend referral to specialist services for diagnosis or interventions). However, I do have some concerns around some areas of the paper. My main concern is around the overall aim of the study – if the aim is to predict current ADHD levels to enable better resource planning and decision making then why not use the actual number of diagnosed cases. It does not make sense to me to predict something when you already have data on the answer. For healthcare resource planning, a model would need to be built to predict future levels of ADHD, rather than current levels. For school resourcing, it may be better if pupils at risk received a diagnosis so that appropriate support could be put in place – which makes me think the model might be more useful making predictions at the individual level. However, I would worry that the model is not accurate enough to be used to recommend individual level interventions. For this reason, I do not currently see the use case for the model. Some additional specific comments are below: In the introduction you mention that ADHD affects an estimated 5.3% of people globally – but it seems to be much lower in this data. Do you have any explanation for this? I felt the methodology section could be clearer. It was only by visiting the CRIS web site that I understood what had been done around the linkage. It was not clear from the ethics section that the ethics approval covers the data linkage – I have a particular concern, as the data would cover children and mental health services, so this needs to be very clear. The ethics statement states that anonymised data can be used for the research, but
---

	earlier it is stated that pseudonymised data was used – this should be clarified as to whether anonymised or pseudonymised data was available. It would be useful to mention what hyperparameters were tuned (particularly as often there are none tuned for logistic regression). I was not sure from the text how the boosting and bagging had been applied to the trained models. In Table 1 it would be useful to know the min and max for EYFSTP and KSI. The bottom half of table 1 could be improved (e.g. moving n and % as column headers). Given that 10-fold cross validation was used to tune the hyperparameters, it wasn't clear to me what the 12.5% validation set was used for. On pg. 7 line 41 it states that the validation set was used for hyperparameter tuning, but was the 10 fold CV not performed on the training set? Why were these specific machine learning algorithms used? It was not clear to me how the sub sampling techniques were applied, or why these specific techniques were used. In particular, was subsampling carried out within the cross validation, and if not, why not? I assume that only the training data was subsampled. How many subsamples were used to calculate the average AUC, and why? I would also be inclined to try upsampling / SMOTE. It was not clear from the text why these specific groupings were used in the fair preprocessing weighting. For example, why not weight by ethnicity and deprivation, or gender and ethnicity – I think this needs more analysis to justify the specific groupings that are used. If the assumption that different ethnicities / language groups have the same underlying population rate of ADHD is correct, then it appears that Asian people are also under represented in the clinical cohort – why was weighting not included here as well? When comparing cohorts it would be useful to state whether the differences are statistically significant. pg. 15 line 35 – It isn't clear how the class imbalance would lead to similar levels of performance, given the data was sub-sampled. It would be useful to know how the number of predicted ADHD cases changed for each ethnicity-language group after the data was weighted by the fair pre processing (e.g. by how much did it go up/down for different groups, were there any changes to predictive accuracy across each group). I have some concerns around the reporting of the results. It is common to use methods like permutation importance or SHAP values to calculate measures of variable importance, due to bias in some out of the box measures – did you consider these instead of coefficients / the standard RF variable importance measures. I have a concern that the coefficients of the logistic regression model may be difficult to interpret – for example, there does not seem to be a reference category for the ethnicity dummy variables
--	--

	(Table S4) – I was not certain why this was, but in any case it makes the coefficients difficult to interpret.
--	--

VERSION 1 – AUTHOR RESPONSE

Reviewer: 1 _____

Dr. Yu Shi, Mayo Clinic

Comments to the Author:

The topic has public health significance. Paper is well written. The authors utilized unique resources of school and clinical datasets. I am not familiar with all the machine learning languages but it seemed to be well performed.

I have several questions/comments:

1. It's helpful to present the sensitivity, specificity, PPV, and NPV of the ML models. These metrics can either be computed from the metrics we report, or are already reported under a different name (given the focus of the paper, we have used terminology which is more common in the ML community):

- PPV is also called precision and is reported in Table 2.
- Sensitivity is also called recall and is reported in Table 2.
- Specificity can be computed from accuracy and sensitivity:
 - 1) $Acc = (TP+TN)/sample\ size$: we know the accuracy and sample size so we get TP+TN
 - 2) sensitivity is TP/P : we know P and sensitivity so we get TP and deduce TN
 - 3) specificity is TN/N
- NPV can be computed from the sensitivity, specificity and prevalence (1.16% in the clinical cohort):

$$NPV = \frac{\text{specificity} \times (1 - \text{prevalence})}{\text{specificity} \times (1 - \text{prevalence}) + (1 - \text{sensitivity}) \times \text{prevalence}}$$

Please find these equivalences in the second table below. Moreover, it should be noted that the ROC curve (and thus its corresponding AUC) is a plot of the true positive rate (i.e. sensitivity) against the false positive rate (i.e. 1-specificity) for varying thresholds (i.e. varying values of probability above which the model classifies as positive). It therefore contains richer information than reporting the sensitivity and specificity for a single decision boundary.

	Accuracy		Precision		Recall		F1-score	
	Valid	Test	Valid	Test	Valid	Test	Valid	Test
LR	0.862	0.900	0.985	0.987	0.835	0.827	0.900	0.895

RF	0.857	0.860	0.985	0.986	0.816	0.802	0.889	0.880
----	-------	-------	-------	-------	-------	-------	-------	-------

	NPV		PPV		Sensitivity		Specificity	
	Valid	Test	Valid	Test	Valid	Test	Valid	Test
LR	0.873	0.882	0.985	0.987	0.835	0.827	0.890	0.887
RF	0.861	0.896	0.985	0.986	0.816	0.802	0.864	0.902

2. Diagnostic codes on ADHD from clinical charts perform well in identifying true ADHD but are not perfect. It should be mentioned as a limitation. Please see Gruschow 2016 and Shi 2020. Thank you, we have added a comment on this to the Strengths and Limitations section, and added these citations to our manuscript.

3. The work on disparity is very innovative. I don't quite understand if the weighing procedure takes away some of the true differences that are not disparity. For example, male gender is a known risk factor for ADHD diagnosis and probably not a result of discrimination.

Our hypothesis in this paper is that diagnostic and treatment discrepancies between different ethnicities and language groups are due to disparity, and not reflective of genuine underlying differences in ADHD prevalence in these populations (e.g. see Shi et al., 2021, for discussion on the possible mechanisms behind such disparities). We have added some discussion on this stance to the Strengths and Limitations section. To clarify, we did not conduct re-weighting on the basis of gender, and therefore make no assumptions as to the role of gender in differential diagnoses.

4. The author acknowledged that the incidence of ADHD is low in the cohort. Can the author compare the incidence to other published studies?

Thank you for raising this important point; the problem of varying prevalences is something we should have made clearer from the outset, because it is central to the positioning of our paper. We have added a paragraph to the introduction explaining various discrepancies in ADHD prevalence estimates from different sources, and why that necessitates the sort of modelling which we go on to conduct.

Will a higher incidence change the performance of the ML models?

A higher incidence wouldn't normally change the performance of the models, for two reasons: (i) we've reached a good performance of ML models overall, possibly due to sample size; (ii) the fact

that there is class imbalance was acknowledged during training so it doesn't cause the model to systematically predict the majority class. In particular, the models were trained (i) using random subsampling, and (ii) using the weighted option that prevents the model from learning to predict the majority class only.

Reviewer: 2 _____

Dr. Quan Do, Mayo Clinic Rochester

Comments to the Author:

This is an interesting study. I have several comments for the authors. Thank you for the opportunity to review your paper!

Your study population was children (6-7 years old), did you have to seek the permission from the parents to involve participants in this study?

The CRIS data resource, including the linked data reported in this study, has received ethical approval as a dataset for secondary analyses. CRIS operates an opt-out system, where all patients have the ability to opt-out of their anonymised data being used for research. Therefore, opt-in consenting procedures were not required for this study. We have added this clarification to the methods section.

In total, how many features were used to build the model? Did the authors do data engineering, feature selection, feature ranking to select strong predictors?

The model was built and tested on the full set of 68 features. We have added this clarification to the paper: "Once all models were tested on the full set of 68 features, we refined the models to improve accuracy, reduce overfitting and increase potential for translation into busy clinical settings."

Have you checked about outlier/outlier removal?

Yes, features were checked for outliers before models were fitted. Once models were fitted, we checked for outlying individuals for the top 10 features taken two by two. This check ensures the model was not doing shortcut learning from outliers in the top features. Outlying individuals weren't a specific issue in this study, and thus we decided not to remove them.

Usually, neural networks (MLP) have better performance than other algorithms, especially Logistic Regression. Do you know why this is not the case in this study? Is your data nonlinear? The dataset is highly complex (size of the cohort, number of features, class imbalance) so it is most likely nonlinear. And yes indeed, in such a scenario, LR usually has lower performance than a neural network due to its heavy parametric assumptions. However, this is only true when the goal is purely predictive. Neural networks are capable of understanding the data in complex ways and predict almost perfectly. However, when doing so, the resulting model isn't explanatory (e.g. it can be learning through a spurious path that isn't causal). In our study, the end goal was to rank features by clinical importance. Our model thus had to reason in an interpretable way to convey this type of information. We thus decided to regularise our MLP heavily to prevent this type of phenomenon (also

known as shortcut learning). For an example of such a problem, see: Geirhos R, Jacobsen JH, Michaelis C, Zemel R, Brendel W, Bethge M, Wichmann FA. Shortcut learning in deep neural networks. Nature Machine Intelligence. 2020 Nov;2(11):665-73.

Missing data (1.4% of data) was deleted. Not sure if you may delete some important cases? There are some imputation methods such as KNN, MICE, or Datawig. Were there any reasons against the use of these imputation methods in this case?

Imputation methods jeopardise our analysis if we aren't aiming to build a purely predictive tool, but understand the underlying data. Further, the imputation method should be approximately as complex as the model. Given that we compare models of varying complexities (from Logistic Regression to MLPs), we would have had to adapt the imputation method to the model, resulting in various datasets. The comparison between models would have been unfair. We have added a comment on this to the Strengths and Limitations section.

It would be better if I can know the changing in accuracy rates through different datasets. Can you please do the 5-fold or 10-fold cross validation and list the differences in accuracy rates of all the used algorithms.

Rather than cross validation, the Confidence Intervals for the AUC, precision, recall and F1-score were built using a similar procedure, namely by bootstrapping 100 times the subject in the validation set. The advantage of bootstrapping over cross validation is that it is non-parametric, which ultimately motivated our choice.

Did you test prediction performance with a completely new dataset?

No, a perspective for future work would be to test prediction performance on data from another catchment area/hospital in the UK (see Strengths and Limitations section). However the linkage of such data is time consuming and has yet to be achieved in other areas, hence the novelty of the existing study.

Reviewer: 3 _____

Dr. Byron Graham, Queens Univ Belfast

Comments to the Author:

Thank you for the opportunity to review this paper. There is definitely merit in predicting ADHD at both the individual level and at aggregate levels. I also found the approach to addressing bias in the underlying data to be an interesting approach, and my feeling is that this is an important consideration especially if this type of tool were used to make predictions at the individual level (E.g. to recommend referral to specialist services for diagnosis or interventions). However, I do have some concerns around some areas of the paper.

My main concern is around the overall aim of the study – if the aim is to predict current ADHD levels to enable better resource planning and decision making then why not use the actual number of

diagnosed cases. It does not make sense to me to predict something when you already have data on the answer. For healthcare resource planning, a model would need to be built to predict future levels of ADHD, rather than current levels. For school resourcing, it may be better if pupils at risk received a diagnosis so that appropriate support could be put in place – which makes me think the model might be more useful making predictions at the individual level. However, I would worry that the model is not accurate enough to be used to recommend individual level interventions. For this reason, I do not currently see the use case for the model.

Thank you for highlighting this. We have strengthened our justification for the model in the introduction. This links closely to your comment on ADHD prevalence - please see below.

Some additional specific comments are below:

In the introduction you mention that ADHD affects an estimated 5.3% of people globally – but it seems to be much lower in this data. Do you have any explanation for this?

Thank you for raising this important point; the problem of varying prevalences is something we should have made clearer from the outset, because it is central to the positioning of our paper. We have added a paragraph to the introduction explaining various discrepancies in ADHD prevalence estimates from different sources, and why that necessitates the sort of modelling which we go on to conduct. This hopefully also helps clarify the use case for the model, as per your main concern above.

I felt the methodology section could be clearer. It was only by visiting the CRIS web site that I understood what had been done around the linkage.

Thank you for this, we agree our description of the linkage was a little sparse. We have added some more detail to the study design section, and signposted readers to the papers which describe the linkage in full.

It was not clear from the ethics section that the ethics approval covers the data linkage – I have a particular concern, as the data would cover children and mental health services, so this needs to be very clear. The ethics statement states that anonymised data can be used for the research, but earlier it is stated that pseudonymised data was used – this should be clarified as to whether anonymised or pseudonymised data was available.

Thank you for spotting this, we have updated the ethics section with the relevant statement that covers the data linkage. We have also updated the paper to use the term ‘anonymised’ consistently throughout.

It would be useful to mention what hyperparameters were tuned (particularly as often there are none tuned for logistic regression).

Logistic Regression: solver

Random Forest: n_estimators, max_depth, max_features, bootstrap, criterion, min_samples_split

SVM: C, cache_size, degree, gamma, kernel, max_iter, tol=0.001

Gaussian Bayes: no tuning

MLP: weight_decay, batch_size, dropout_rate, number of hidden layers

We have added this information to the supplementary material (Table S4).

I was not sure from the text how the boosting and bagging had been applied to the trained models.

We've deleted this sentence from the manuscript. We attempted to apply bagging and boosting to tuned models to improve them further, but this was not successful, so we have removed reference to them to avoid confusion.

In Table 1 it would be useful to know the min and max for EYFSTP and KSI.

We have added this information to Table 1.

The bottom half of table 1 could be improved (e.g. moving n and % as column headers).

Thank you for this suggestion, we have amended Table 1 accordingly.

Given that 10-fold cross validation was used to tune the hyperparameters, it wasn't clear to me what the 12.5% validation set was used for. On pg. 7 line 41 it states that the validation set was used for hyperparameter tuning, but was the 10 fold CV not performed on the training set?

The 10 fold CV was performed on the validation set for hyperparameter tuning: "The training set was used to train the algorithms, while the validation set was used for model refinement (e.g., hyperparameter tuning) and comparison." We have also now clarified this in the section on tuning: "For LR and RF, tuning of the hyperparameters was made using grid search with 10-fold cross validation on the validation set (full details available by request to the author)."

Why were these specific machine learning algorithms used?

We chose this set of machine learning algorithms because they are state-of-the-art techniques and are representative of the wide spectrum of machine learning methods: an ensemble method (RF), a kernel-based method (SVM), a Bayesian method (Naive Bayes), and a neural network (MLP). We have added a comment on this to the Strengths and Limitations section.

It was not clear to me how the sub sampling techniques were applied, or why these specific techniques were used. In particular, was subsampling carried out within the cross validation, and if not, why not? I assume that only the training data was subsampled. How many subsamples were used to calculate the average AUC, and why? I would also be inclined to try upsampling / SMOTE.

Subsampling was used to overcome class imbalance, where the model can achieve good accuracy by predicting the majority class systematically. We tried using upsampling but the results weren't as

good. Furthermore, upsampling has been criticised when used in a fairness context (see Boratto L, Fenu G, Marras M. Interplay between upsampling and regularization for provider fairness in recommender systems. *User Modeling and User-Adapted Interaction*. 2021 Jul;31(3):421-55.)

It was not clear from the text why these specific groupings were used in the fair preprocessing weighting. For example, why not weight by ethnicity and deprivation, or gender and ethnicity – I think this needs more analysis to justify the specific groupings that are used. If the assumption that different ethnicities / language groups have the same underlying population rate of ADHD is correct, then it appears that Asian people are also under represented in the clinical cohort – why was weighting not included here as well?

Our hypothesis in this paper is that diagnostic and treatment discrepancies between different ethnicities and language groups are due to disparity, and not reflective of genuine underlying differences in ADHD prevalence in these populations (e.g. see Shi et al., 2021, for discussion on the possible mechanisms behind such disparities). We therefore wanted to illustrate a potential method for mitigating this bias. We acknowledge that this is a preliminary step, and have added some discussion on this stance to the Strengths and Limitations section.

We did not conduct re-weighting on the basis of gender or deprivation, and therefore make no assumptions as to their role in differential diagnoses. This could be an area for future work. To clarify, Asian people were included in the non-White ethnic group which was used for re-weighting, so their under-representation in the clinical cohort was reflected in that procedure, albeit not at a granular level. Again, this would be an area for future work, and we have added this to the Strengths and Limitations section.

When comparing cohorts it would be useful to state whether the differences are statistically significant.

Given the ongoing replication crisis, we are very wary of running *post hoc* statistical tests when we did not design the study to specifically answer the corresponding directional hypotheses and research questions that would accompany each variable. Indeed for some variables (e.g. Summer birth), we would be predicting the null hypothesis, which runs counter to the purpose of hypothesis testing. For this reason, we would prefer to allow the reader to draw such comparisons descriptively.

pg. 15 line 35 – It isn't clear how the class imbalance would lead to similar levels of performance, given the data was sub-sampled.

We have removed this sentence. We wanted to explain the similar performances of the models while being modest about our results, but indeed the subsampling has likely prevented our models from learning the majority class only.

It would be useful to know how the number of predicted ADHD cases changed for each ethnicity-language group after the data was weighted by the fair pre processing (e.g. by how much did it go up/down for different groups, were there any changes to predictive accuracy across each group).

We've reported on the ratios as they are the most informative, and to avoid redundancy. Absolute values can be recovered from the numbers of true positive and true negative cases in each category (either language or ethnicity).

I have some concerns around the reporting of the results. It is common to use methods like permutation importance or SHAP values to calculate measures of variable importance, due to bias in some out of the box measures – did you consider these instead of coefficients / the standard RF variable importance measures. I have a concern that the coefficients of the logistic regression model may be difficult to interpret – for example, there does not seem to be a reference category for the ethnicity dummy variables (Table S4) – I was not certain why this was, but in any case it makes the coefficients difficult to interpret.

Shapley values have been considered as a feature attribution method. However they present two considerable challenges.

The first challenge is the time complexity of their computation, especially on such a large dataset. Shapley values are a sum on feature coalitions, therefore the computation times increase exponentially with the number of features (2^d , with $d=67$ here).

The second challenge is the choice of a reference distribution. This characteristic of SHAP has been heavily debated. A marginal reference distribution breaks the correlation between features. A conditional reference distribution can give misleading interpretations as it violates the “Dummy” Property of Shapley values (see Sundararajan M, Najmi A. The many Shapley values for model explanation. In International conference on machine learning 2020 Nov 21 (pp. 9269-9278). PMLR.)

For the reweighing procedure, we dichotomised ethnicity as White/non-White, and language as English/non-English. Therefore the coefficients can be interpreted as their importance after reweighing was done to counteract potential disparity between white/non-white subjects.

VERSION 2 – REVIEW

REVIEWER	Shi, Yu Mayo Clinic
REVIEW RETURNED	31-May-2022
GENERAL COMMENTS	Thank you for revising the manuscript! I don't have further questions.
REVIEWER	Do, Quan Mayo Clinic Rochester
REVIEW RETURNED	19-Jun-2022
GENERAL COMMENTS	This is an interesting topic and the paper was well-written. The methods were described carefully with details. Results are limitations were discussed.